# Assessment of Main Cereal Crop Trade Impacts on Water and Land Security in Iraq

**Salam Hussein Ewaid** [1] , **Salwan Ali Abed** [2] **and Nadhir Al-Ansari** [3],*

1    Technical Institute of Shatra, Southern Technical University, Basra 61001, Iraq; salamalhelali@stu.edu.iq
2    College of Science, University of Al-Qadisiyah, Diwaniyah 58001, Iraq; salwan.abed@qu.edu.iq
3    Department of Civil, Environmental and Natural Resources Engineering, Luleå University of Technology, 97187 Luleå, Sweden
*    Correspondence: nadhir.alansari@ltu.se; Tel.: +46-(0)920-491858

**Abstract:** Growing populations, socio-economic development, the pollution of rivers, and the withdrawal of fresh water are all signs of increasing water scarcity, and with 85% of global use, agriculture is the biggest freshwater user. The water footprint (WF) and virtual water (VW) are concepts used recently for freshwater resources assessment. The WF reflects how much, when and where the water was used whereas VW reveals the volume of water embedded in goods when traded. The first goal of this research is to determine the WF per ton and the WF of production ($Mm^3$/yr) of wheat, barley, rice, and maize in Iraq. The second goal is estimating the quantities of the 4 main cereal crops imported into Iraq and assessing the impact on reducing WF and land savings for 10 years from 2007 to 2016. The results showed that the WF per ton was 1736, 1769, 3694, 2238 $m^3$/ton and the WF of production was 5271, 1475, 997, 820 $Mm^3$/yr for wheat, barley, rice, and maize, respectively. The median total VW imported was 4408 $Mm^3$/yr, the largest volume was 3478 $Mm^3$/yr from wheat, and Iraq saved about 2676 $Mm^3$ of irrigated water and 1,239,539 M ha of land by importing crops every year during 2007–2016. The study revealed the significance of better irrigation management methods to decrease the WF through a selection of crops that need less water and cultivation in rain-fed areas, as well as the use of cereal import to conserve scarce water resources, which is crucial both in terms of water resource management and preservation of the environment. The results of this research could be used as a guideline for better water management practices in Iraq and can provide helpful data for both stakeholders and policymakers.

**Keywords:** cereal crops; virtual water trade; footprint; Iraq

## 1. Introduction

Over the past few decades, there have been growing demands for freshwater because of widespread population growth, economic activity, and irrigation. The need to use water subsequently has become one of the world's major environmental issues [1].

Due to water shortages, low rainfall, and high evapotranspiration rate, the situation is more severe in arid and semiarid areas. Since agricultural water consumption for many regions is the largest fraction of freshwater use, understanding how water can be used and trading of specific crops is of great importance in reducing water shortages [2].

Iraq is a country facing major environmental, security, political and economic challenges that are interconnected, and 85% of its renewable freshwater resources are consumed by the agricultural sector [3].

Agricultural production in Iraq is divided into rain-fed agriculture, primarily in the north, and irrigated farming, primarily in the Tigris and Euphrates plains. The most rapidly growing rain-fed

crops in the north are wheat and barley, the same winter crops that are cultivated in the irrigated plain in the middle and south of the country, representing the main part of the cereal production. The main irrigated summer crops are rice and maize [4].

Climate change effects, rising temperatures, severe droughts, decreasing rainfall, desertification, and salinization have undermined Iraq's agricultural sector [5]. In addition, there is an expected rise in water demand in Iraq from the present point of 16% to 51% in 2050 owing to increasing temperature, reduced rainfall and the threat of water scarcity based on two declining rivers, the Euphrates and the Tigris, which are expected to decline by 50% and 25% respectively by 2025 [6,7].

Bread wheat (*Triticum aestivum*) is a major part of the daily diet in Iraq. Consistent with the Iraqi Central Statistical Organization yearly statistics [8], winter wheat is Iraq's main crop, accounting for approximately 70% of total cereal production. The areas for the production of wheat are rain-fed, irrigated and mixed, located in all the country, where the climatic conditions favor the cultivation needs. The wheat is grown in mid-November and the harvest period starts from the mid of April until late May [9]. Wheat production is expected to increase in Iraq in 2019 up to 10.34% due to the high winter precipitation in all wheat cultivation areas [10].

Barley (*Hordeum vulgare*) after wheat, maize, and rice is the world's fourth most important cereal crop. It is mainly produced for animal feed, malt products, and human food supply. Barley is generally more competitive than wheat and other grains and its yield is less affected by seasonal variations [11]. Barley after wheat is the second-largest cereal crop in Iraq, and its growing season is similar to wheat and is commonly between November and May [9].

Similar to the analysis for wheat, barley production can be divided into three regions; northern, central and southern Iraq. Production in central and southern Iraq is irrigated (due to low rainfall), while production in northern Iraq is dependent on rainfall [4,12].

Rice (*Oryzae sativa*) is the main food for more than half of human beings around the world. Approximately 480 million tons of milled rice is produced each year. China and India are the global leaders in rice consumption, amounting to 50% of global consumption [13].

In Iraq, rice is the third most popular crop after wheat and barley in terms of planted area and production, but Iraq is a rice-importing country, and its rice production is not enough to meet the needs of its population [8].

The production of rice as the main summer crop in Iraq is mainly in the Middle Euphrates (Najaf and Qadisiyah provinces) in central and southern Iraq and the growing season starts in June/July and ends in October/November [14].

Maize (*Zea mays*) is an important cereal for human food and animal feed, used in worldwide starch production and grown for grain and forage, and it is the main component of animal forage in Iraq, produced and imported into the country every year [15]. The maize-producing areas are located in all Iraqi provinces [8].

Allen et al. [16] first mentioned the term virtual water (VW) to support the idea that water-scarce regions could save their water resources for municipal needs by importing more food rather than producing food themselves [17,18]. During the past 20 years, the concept of VW has gained growing interest to describe the water contained in a product required for processing, packaging, and shipping the commodity to consumers [19].

The water footprint (WF) shows the total water consumption as measured for individual consumers, populations, nations or companies [20].

Virtual water and water footprint have helped shed light on agriculture's role in global water management and the role of trade in water poverty alleviation [21]. There are three footprints or virtual water components: green water (rainwater used to grow crops), blue water (irrigation water) and greywater (wastewater generated by different activities) [16].

Taking advantage of the knowledge of the water footprint and virtual water trading has proved to be a water-saving choice and for countries at the edge of their freshwater resource like Iraq, WF and

VW trade can supplement endogenous water with exogenous sources and form the basis for potential water stress mitigation [22].

Food trade in water-scarce areas is a significant component of food security, it contributes to water conservation or export through the VW trade, referring to the trade-in water incorporated in food products [23]. Virtual water ideas and quantitative estimates can assist in realistically evaluating water scarcity for each nation, projecting future water demand for food supply, raising government awareness of water and identifying water-waste procedures in manufacturing [17].

For water-scarce nations, achieving water security by importing water-intensive products could be a more attractive alternative compared to all water-intensive products produced locally [24]. It is worth noting that in ensuring the food requirement, rain-fed crops play a crucial role but total internal use of water involves both the use of green and blue water [25].

In a number of previous studies, the WF and the respective savings generated by the trade in agricultural goods were quantified. Fader et al. [26] showed that crop trade saves 263 billion $m^3$/yr of water worldwide, representing 3.5% of annual rainfall cropland. In particular, water-scarce nations such as Mexico and China, but also Japan and the Netherlands, by importing products (25–73 billion $m^3$) of water, saved big quantities of water because they needed comparatively big quantities of water to manufacture the products they imported.

The study of Biewald et al. [27] showed that blue water savings from international trade can provide enormous advantages to water-scarce areas; for instance, worldwide food trade has saved 17 billion $m^3$/year of blue water. Many studies have shown that efficient VW imports can decrease water consumption for national food production in importing nations and help relieve water stress in water-scarce areas such as Iraq [24].

A study conducted by [28] reported that the WF of production for wheat, barley and maize in Iran for 2006–2012 were 36,777, 7975 and 3744 $Mm^3$ respectively.

Lee et al. [22] analyzed the impact of food trade on water-land savings in the Middle East and North Africa and concluded that the region saved important amounts of water and land-based on the importation of four main crops; barley, maize, rice, and wheat, between 2000 and 2012, although food self-sufficiency remains limited. Egypt, for instance, imported 8.3 million tons of wheat per year, resulting in 7.5 billion $m^3$ of irrigation water and 1.3 million ha of land savings.

Muratoglu [29] analyzed the blue and green WF of the upper Tigris River basin in Turkey for the years 2010–2018, and found that the annual WF was 7.2 $Gm^3$/yr corresponding 1748 $m^3$/cap/yr, crop production has 79% of total WF, and wheat is responsible for 45% of water use of all crops.

Iraq's critical water shortage will reach serious rates by 2050; furthermore, if the population rises quickly and urbanization continues fast, water availability in Iraq could decrease and water shortages will accelerate Iraq's desertification rate with a greater freshwater deficit [7,30,31].

Accordingly, in relation to enhancing food security in Iraq, the food trade can be considered as the most significant factor in saving local water resources and reducing water stress.

The objective of this research is to address the role and effect of the cereals trade to Iraq and its impacts on Iraq's water and land savings.

## 2. Data and Methods

### 2.1. Study Area and Data Collection

Iraq is located at longitude (38°48′ E), latitude (29°37′ N), covers an area of 438,317 $km^2$ in the Middle East which characterized by its water shortage problem (Figure 1). Iraqi land can be divided into 4 regions: 1—The mountain region in the north occupied 5% of the total area. 2—Hills region represents 15%. 3—The alluvial plain is restricted between the Tigris and Euphrates and occupies 20%. 4—Jazeera and Western Plateau forms 60% of the total area [31].

Approximately 11.5 million ha (26%) of the country's total region is cultivable and the total cultivated region is about 6 million ha, of which nearly 50% in rain-fed conditions in northern

Iraq [32]. The total yearly water withdrawal in Iraq is approximately 42.8 billion m$^3$/yr, for agricultural (85%), domestic (7%) and industrial (8%) purposes [33]. The climate is a subtropical semi-arid type. The average daily temperature during winter is around 16 °C. It is very hot in summer with an average temperature above 45 °C. The total annual average rainfall is 213 mm per year. The rainy season begins in October and ends in April [34]. The study region contains all the Iraqi provinces excluding the 3 provinces of Kurdistan (Duhok, Arbil and Sulaymaniyah) due to lack of data.

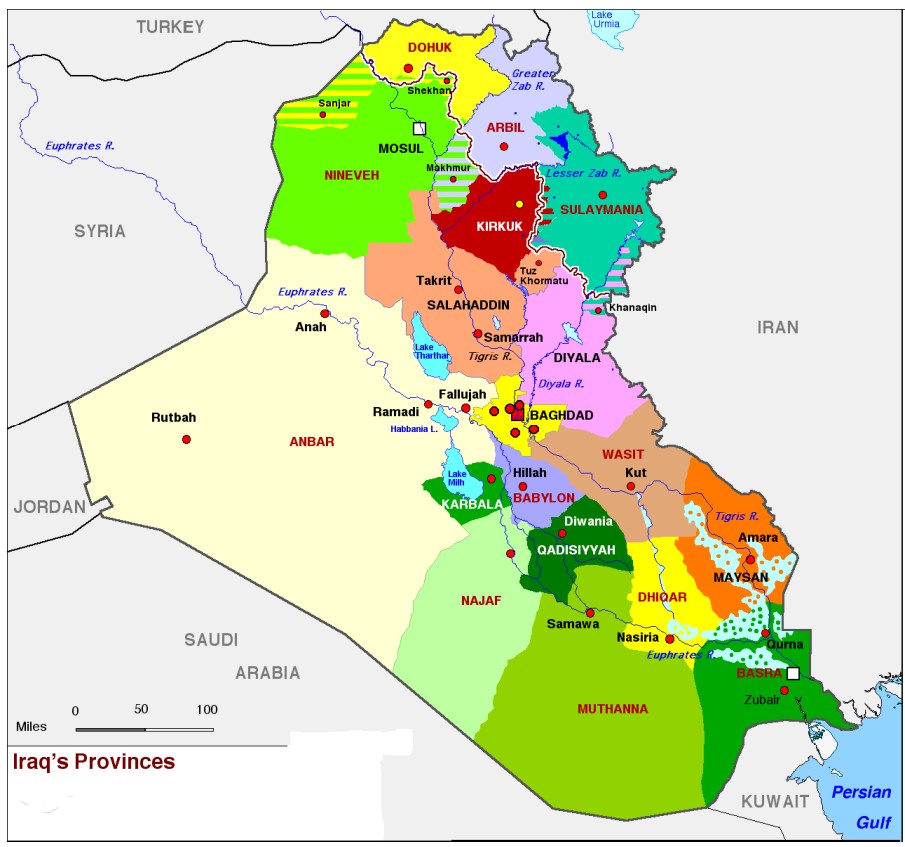

**Figure 1.** Iraqi map (the study area).

The amount of water required to grow a certain crop is needed to calculate the water footprint (WF, m$^3$/ton) content of the cultivated crop. The calculation is based on accumulated evapotranspiration during growing crop periods which is referred to as the crop water requirement (CWR, m$^3$/ha).

The main variables affecting CWR quantity are precipitation, temperature, air humidity, hours of sunlight, wind speed, type of crop, soil conditions and time of cultivation. The United Nations Food and Agriculture Organization (FAO) recommended that the standard Penman–Monteith equation is often used to calculate the evapotranspiration [35].

Using the United Nations Food and Agriculture Organization (FAO) decision support system CROPWAT (Crop Water Requirement), plant evapotranspiration (ETc) was determined following the FAO method [16]. It can calculate the irrigation requirements and CWR based on climate, soil, and crop data. The climate data includes the average min and max temperature, sunshine hours, precipitation, wind speed, and relative humidity. Climatic data for 30 years (1971–2000) was received from 12 Iraqi meteorological stations, obtained by the CLIMWAT 2.0 which is a climatic database to be used with the CROPWAT model [36–38].

The agricultural data includes crop production, cultivated area, planting, and harvesting dates were taken from the Iraqi Ministry of Agriculture [15].

Since Iraq has been considered as a non-reporting country for trade data (i.e., no official trade data for Iraq have been provided to the UN Statistics Division under COMTRADE (Global international

trade data repository) https://comtrade.un.org/data where FAO get to trade for all countries from) mirror statistics have been used by FAO. The mirror data for Iraq was found in FAOSTAT http://www.fao.org/faostat/en/#data/TM: the 10 years of trade data of the study crops from 2007 to 2016 were obtained from this site.

*2.2. Methods*

The constituents of crop water footprint (WF) are the types of water used in the crop cultivation process, including green water and blue water. The green water is equal to the total WF of the crop under a rain-fed scenario. The WF of a crop can be computed by the method of Hoekstra et al. [18,36] as follows:

$$WF = \frac{CWU}{Y} \tag{1}$$

where, WF is the water footprint content of the crop by (m$^3$/ton), Y is the crop yield per unit area (ton/ha) and CWU is the crop water use (m$^3$/ha). The CWU is calculated from all the crop evapotranspiration (ET$_c$) during the growing period:

$$CWU = 10 \times \sum_{d=1}^{Igp} ET_c \tag{2}$$

where Igp is the duration of the period of growth, factor 10 is used to convert water depth (mm) into water volume (m$^3$/ha); the calculation of evapotranspiration during the crop-growing period (mm) was as follows:

$$ET_c = K_c \times ET_0 \tag{3}$$

ET$_0$ (mm/dec) is the reference crop evapotranspiration calculated by the FAO Penman–Monteith equation and K$_c$ is the dimensionless crop coefficient.

The blue and green WF for a crop is calculated as the blue and green water in the crop water use CWU divided by the amount of the crop yield per unit area Y (ton/ha). The values for blue and green water in CWU are equal to the daily accumulation of the evapotranspiration over the period of crop growing:

$$WF_{blue} = \frac{CWU_{blue}}{Y} = 10 \times \frac{ET_{blue}}{Y} \tag{4}$$

$$WF_{green} = \frac{CWU_{green}}{Y} = 10 \times \frac{ET_{green}}{Y} \tag{5}$$

where WF$_{blue}$ and WF$_{green}$ are the blue and green components, respectively of the WF for the crop (m$^3$/ton), CWU$_{blue}$ and CWU$_{green}$ are the blue and green water consumed during the period of the crop-growing (m$^3$/ha), ET$_{blue}$ and ET$_{green}$ are the blue and green water evapotranspiration during the period of the crop-growing (mm). ET$_{blue}$ and ET$_{green}$ can be assessed using the FAO CROPWAT model and the CLIMWAT database:

$$ET_{green} = \min(ET_c, \text{ R. eff.}) \tag{6}$$

$$ET_{blue} = \max(0, ET_c - \text{ R. eff.}) \tag{7}$$

where R.eff. is the effective rainfall during the period of the crop-growing (mm). The rain data in CLIMWAT was used to calculate the effective rainfall (Eff. rain) using the United States Department of Agriculture (USDA) South Carolina method [16].

The CWU, R.eff. and irrigation requirements (IR) of wheat, barley, rice, and maize were calculated for each province. Then, according to the yield of the four crops in the producing provinces, the blue and green WF was calculated. In order to further analyze the regional differences of WF for the four crops, meteorological and yield data from the producing provinces were used to calculate it.

The total WF of crop production (Mm$^3$/yr) is the volume of water used to develop all the country's production of that particular crop for a year and is the summation of the blue and green WF.

$$\text{WF}_{\text{green}}\left(\text{Mm}^3/\text{yr}\right) = \text{WF}_{\text{green}}\left(\text{m}^3/\text{ton}\right) \times \text{production (ton)} \tag{8}$$

$$\text{WF}_{\text{blue}}\left(\text{Mm}^3/\text{yr}\right) = \text{WF}_{\text{blue}}\left(\text{m}^3/\text{ton}\right) \times \text{production (ton)} \tag{9}$$

*2.3. Water Footprint-Reducing and Land-Saving Methodology*

The water footprint of trade (WFT) is water integrated into international trade, the key factors for quantifying it are trade information and water footprint (m$^3$/ton), which is the amount of water used to generate one ton of the crop; the WFT is calculated as follows:

$$\text{WFT}\left(\text{Mm}^3\right) = \text{CT (ton)} \times \text{WF}\left(\text{m}^3/\text{ton}\right) \tag{10}$$

where WFT is the water footprint of trade quantity, CT represents the crop amount imported to the country, and WF denotes the water footprint of the crop in the country.

Importing crops can reduce the local WF and affect any country's savings in water and land use. Hence, poor trade management can cause water and land shortages. Therefore, in this study, the water and land requirements were analyzed to produce as much crop as it is imported. As such, water and land savings indicated the requirements for resources required by the shift from crop imports to local production. National water and land savings indicated the amount of blue water and land required to replace imported crops with local production. Therefore, it was calculated as follows, adjusted to keep in mind Iraqi imports only and not exporting crops abroad [22]:

$$\text{water saving (WS)}\left(\text{m}^3\right) = \text{imported crop quantity (ton)} \times \text{blue WF}\left(\text{m}^3/\text{ton}\right) \tag{11}$$

$$\text{land saving (LS)(ha)} = \text{imported crop quantity(ton)} \times \frac{\text{land (ha)}}{\text{production (ton)}}, \tag{12}$$

## 3. Results and Discussion

*3.1. Water Footprint (m$^3$ per ton) of the Four Crops*

Table 1 shows the main results of this study. The averages annual WF for wheat, barley, rice, and maize are 1736, 1769, 3669 and 2238 m$^3$/ton, respectively, for 2007–2016. The green WF is around 891, 1143, 10, 69 m$^3$/ton respectively which corresponds 51.3%, 64.7%, 0.3%, 3.08% respectively of the total WF. The difference is clear and significant among winter crops (wheat and barley) and summer crops (rice and maize) which consume much blue water, Figures 2 and 3.

**Table 1.** The average annual data of wheat, barley rice and maize production, cultivated area, yield, crop water use, WF of production and WF per ton in the 15 Iraqi provinces (2007–2016) [11,15].

| Crop | Planted Area | Production | | Crop Water Use (m$^3$/ha) | | | WF of Production (Mm$^3$/yr) | | | Water Footprint (m$^3$/ton) | | |
|---|---|---|---|---|---|---|---|---|---|---|---|---|
| | ha | ton/yr | ton/ha | Green | Blue | Total | Green | Blue | Total | Green | Blue | Total |
| Wheat | 1,786,235 | 3,036,882 | 1.700 | 1257 | 1181 | 2438 | 2,707,073,151 | 2,563,931,419 | 5,271,004,570 | 891 | 844 | 1736 |
| Barley | 623,666 | 833,815 | 1.336 | 1203 | 1119 | 2322 | 953,170,654 | 522,128,613 | 1,475,299,267 | 1143 | 626 | 1769 |
| Rice | 68,897 | 269,970 | 3.918 | 717 | 13.127 | 13,199 | 2,692,008 | 994,531,633 | 997,223,641 | 10 | 3684 | 3694 |
| Maize | 123,678 | 366,523 | 2.963 | 288 | 9.096 | 9384 | 25,292,837 | 794,944,139 | 820,236,976 | 69 | 2169 | 2238 |
| Total | 2,602,476 | | | 3465 | 2322 | 27,343 | 3,688,228,650 | 4,875,535,804 | 8,563,764,454 | | | |

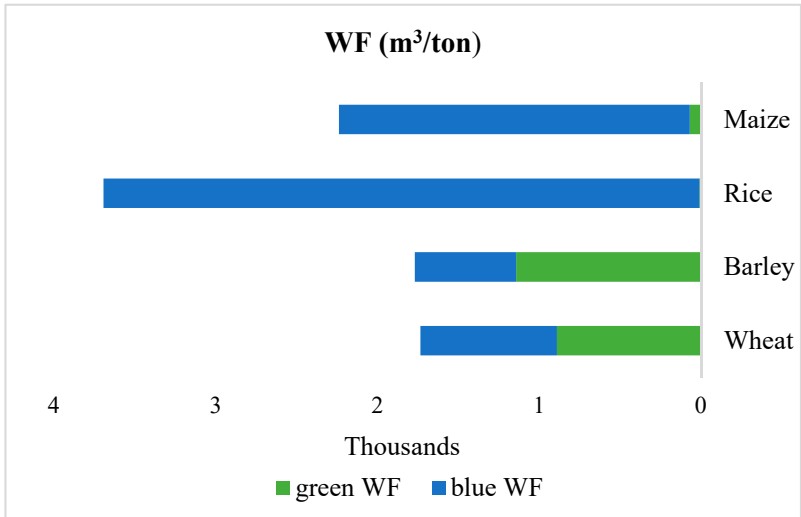

**Figure 2.** The annual components of water footprint (WF, m$^3$/ton) of the 4 crops for 2007–2016.

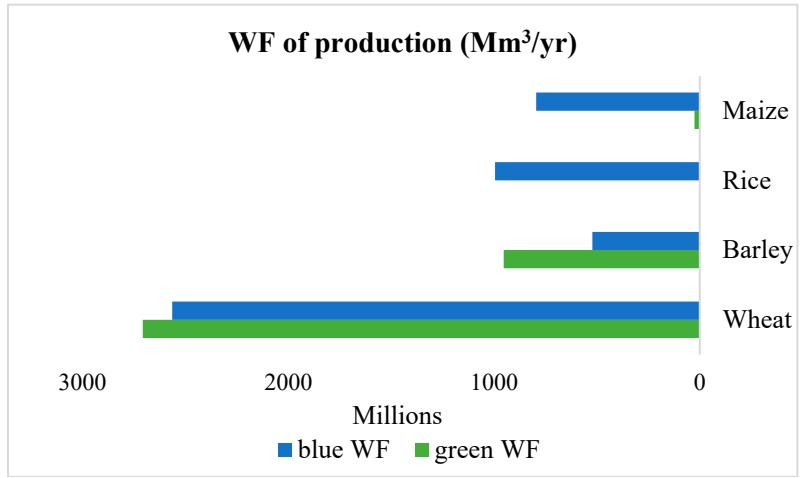

**Figure 3.** The components of the annual WF of production (Mm$^3$/yr) of the 4 crops for 2007–2016.

*3.2. The Water Footprint of Crop Production*

The WF for crop production (Mm$^3$/yr) which means all the water needed to produce the crop in the country within a year was calculated for the 4 crops as listed in Table 1 and is illustrated in Figure 3. Wheat production has the highest WF, wheat green WF of production was 2,707,073,151 Mm$^3$/yr (51.35%), blue was 2,563,931,419 (48.65%) Mm$^3$/yr, and the total wheat WF of production was 5,271,004,570 Mm$^3$/yr; other crops have lower WF. The average total annual WF of the 4 crops within the 15 Iraqi provinces for 2007–2016 is calculated to be 8,563,764,454 Mm$^3$/yr (Table 1).

Table 1 and Figure 4, show the annual average data on wheat, barley rice, and maize cultivated area, production, and import in the 15 Iraqi cereal-producing provinces during 2007–2016.

There is obviously a fluctuation in the quantities of area cultivated, produced and imported crops for the four crops annually. Many studies report that a lack of water is the cause of the fluctuation in the cultivated area, production, and imports [3,12,39].

In the three northern Iraqi provinces of Nineveh, Kirkuk, and Saladin, full reliance is placed on rainwater to irrigate wheat and barley crops in the winter. In the rest of the central and southern provinces, more reliance is placed on river water to irrigate all crops [6].

Taking the average cropped area of 10 years, a total area of 1,786,235 ha of wheat was grown, of which 624,213 ha was in the northern provinces. This would show that at least 35% of the wheat area is rain-fed, and so 65% of the wheat is probably to be at least partly irrigated, this would represent an

area of 1,162,022 million ha of irrigated wheat, most of which would grow in the middle and south of the country (Table 1, Figures 4–6).

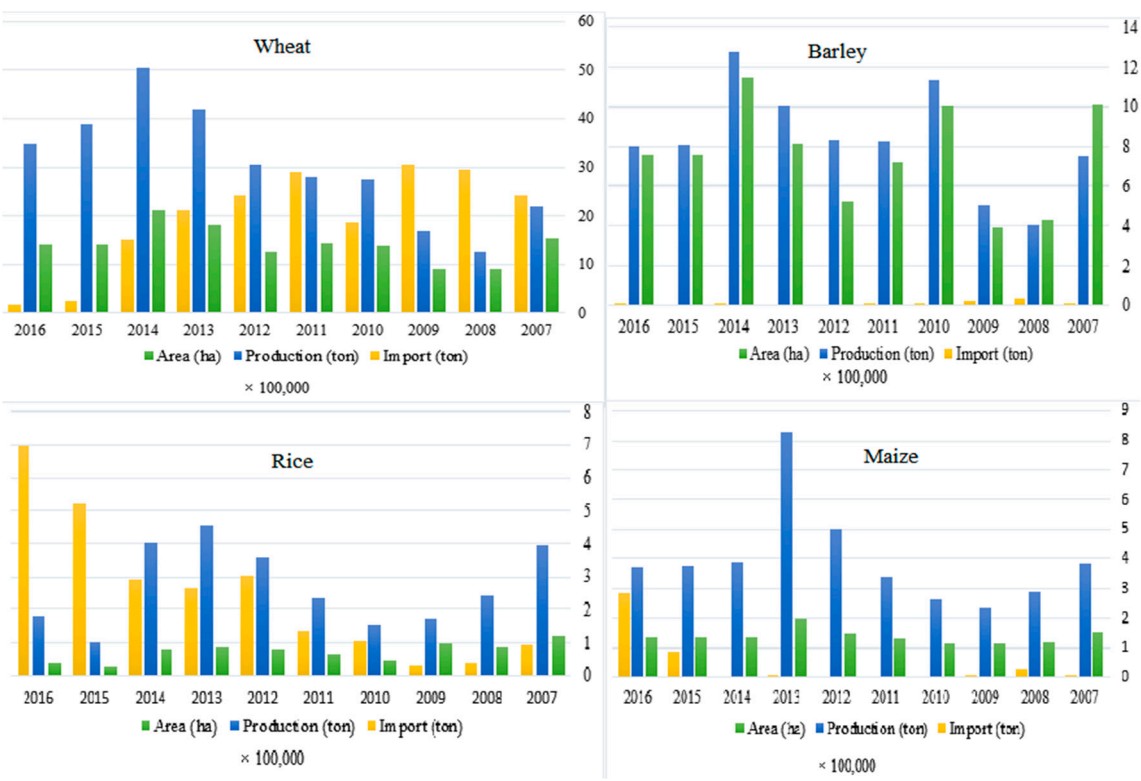

**Figure 4.** The annual average data of cultivated area (ha), production (ton), and import (ton) for wheat, barley, rice, and maize crops, 2007–2016.

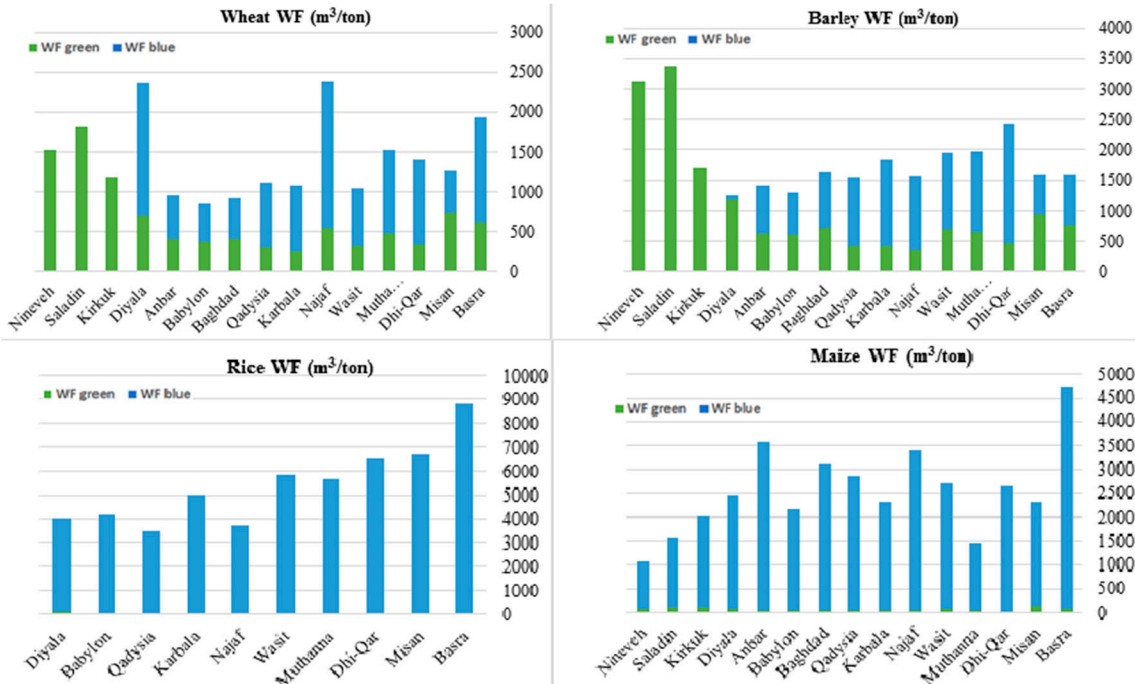

**Figure 5.** The average of the WF components (m³/ton) of the 4 crops 2007–2016.

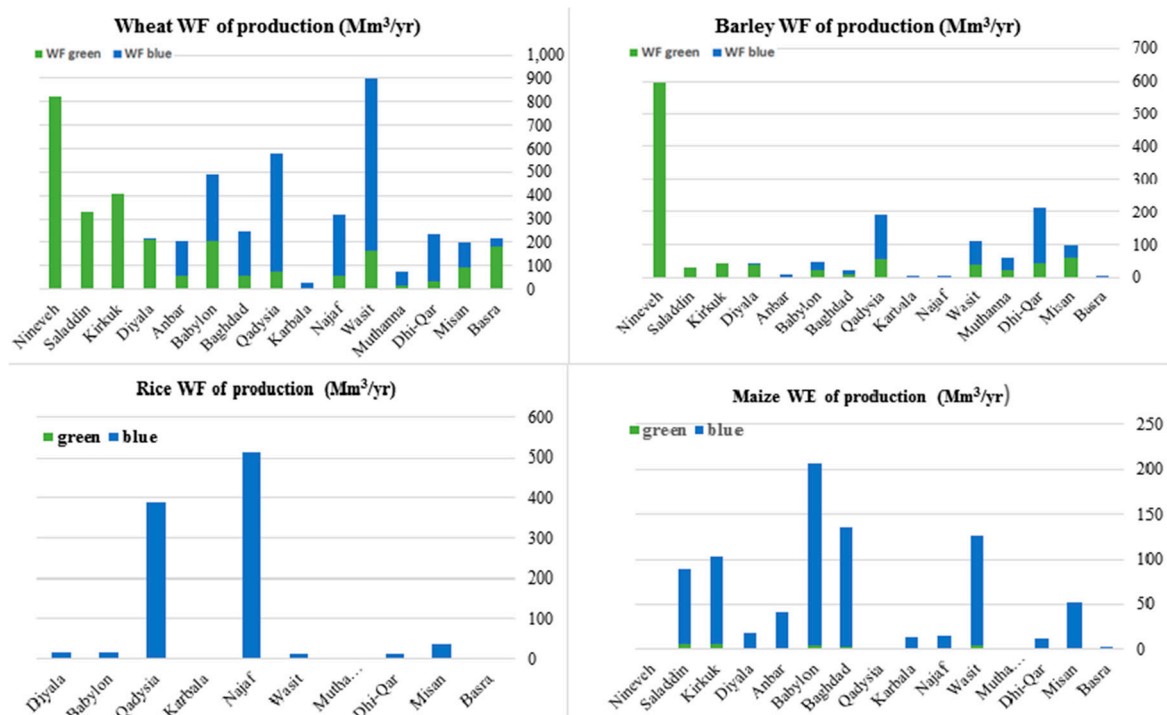

**Figure 6.** The average WF of production components (Mm$^3$/yr) of the 4 crops 2007–2016.

For barley, in the northern three provinces, there was more reliance placed on rainwater, there were 409,908 tons (49%) of the total 833,815 tons produced in the 296,510 ha (48%) from 622,666 ha total area (Table 1 and Figure 4). Rice production in Iraq as a summer crop, with a big concentration around Najaf province, is primarily cultivated in the irrigated provinces of central and southern Iraq and not cultivated in the north, with an output rate reaching 269,970 ton/year (Table 1 and Figure 4). Production of maize in Iraq as a summer crop of feed is in all 15 provinces, especially Babylon and Wasit in the middle and Kirkuk in the north (Table 1 and Figure 4). It is clear from Table 1 and Figure 4 that Iraq's annual production of the four crops fluctuates and is not sufficient for its local needs due to the amount of rainfall most often. This is compensated by imports from abroad, especially wheat and rice. The detailed WF of the 4 crops is provided in the following paragraphs, Figures 5 and 6.

Wheat is a strategic crop for Iraq. The cultivated area, production, and import is the largest among the other crops. To this end, the distribution of water resources in wheat production has a profound effect on the consumption of domestic freshwater and needs to be handled with greater accuracy and conservatism [15,39].

Given that Iraq is situated in an arid area of the globe, the findings of this study indicate the significance of improved irrigation management policies to decrease the WF, which is essential both in terms of water resource management and environmental conservation, these findings indicate an alternative (wheat and barley) choice. In the northern provinces (Nineveh, Saladin, and Kirkuk), the proportion of wheat green WF is the only one used for the wheat production, this demonstrates that there are excellent possibilities to improve water productivity and reduce water use in irrigated land by enhancing production in these rain-fed provinces. Shifting attention from reducing the share of blue WF to enhancing the productivity of green WF (the yield per unit of green water use) or the use of supplementary irrigation in rain-fed wheat production schemes (to boost output without significantly raising the blue WF share) appear to be promising options worth further investigation.

Due to negligible precipitation during the summer, irrigation is the only source of water for rice, therefore, crops that are less water consuming and more economically valuable should be cultivated.

### 3.2.1. Wheat

Table 1, Figures 5 and 6 summarize the ten-year average crop water use (CWU), WF per ton (m$^3$/ton) and WF of production (Mm$^3$/yr) components of wheat. The average of the wheat WF per ton was 1736 m$^3$/ton, the green WF average was 981 m$^3$/ton (56%) and the blue WF was 844 m$^3$/ton (44%); however, the blue WF was more than the green WF except in the three Northern provinces (Kirkuk, Saladin, and Nineveh). The green WF of production was 2,707,073,151 Mm$^3$/yr (51.35%), the blue was 2,563,931,419 (48.65%) Mm$^3$/yr and the total WF of production was 5,271,004,570 Mm$^3$/yr. There is no irrigated wheat in the three northern provinces and they have 57.5% of the country's green WF. Najaf, Diyala, and Basra have the largest blue and total WF per ton. Nineveh and Wasit have the largest blue and total WF of production (Figures 5 and 6).

### 3.2.2. Barley

Table 1, Figures 5 and 6 summarize the 10-year average crop water use (CWU), WF per ton (m$^3$/ton) and WF of production (Mm$^3$/yr) components of barley. The average of the WF per ton was 1769 m$^3$/ton, the green WF average was 1143 m$^3$/ton and the blue WF was 626 m$^3$/ton, the blue WF was more than the green WF except in the three Northern provinces (Kirkuk, Saladin, and Nineveh). The green WF of production was 953,170,654 Mm$^3$/yr (64.6%), the blue was 522,128,613 (35.4%) Mm$^3$/yr and the total WF of production was 1,475,299,267 Mm$^3$/yr. There is no irrigated barley in the three northern provinces and they have 45.2% of the country green WF. Saladin and Nineveh have the largest green and total WF per ton and Nineveh has the highest WF of production (Figures 5 and 6).

### 3.2.3. Rice

As a summer crop, rice is not cultivated in the three northern provinces of Iraq and two other provinces (Baghdad and Karbala); it is cultivated in the 10 middle and southern provinces with the highest water WF among the four studied crops. As shown in Table 1, Figures 5 and 6, the average of the WF per ton was 3694 m$^3$/ton, the green WF average was only 10 m$^3$/ton because rice is a summer group and the blue WF was the main component with 3684 m$^3$/ton. The green WF of production was only 2,692,008 Mm$^3$/yr (0.27%), the blue was 994,531,633 (99.73%) Mm$^3$/yr and the total WF of production was 997,223,641 Mm$^3$/yr. Najaf and Qadysia have the largest WF of production because they are the highest productive provinces. Basra has the largest WF per ton because of the hot climate and low production (Figures 5 and 6).

Paddy rice requires more water than any other crop because its fields are cultivated during the hot air of summer under flood conditions. Understanding the requirement for crop water, reliance on blue water and the economic value for a crop grown in a specific season, this creates the fundamental blocks for planning an economically and environmentally sustainable suitable crop pattern. Efficiency in water allocation can be increased through spatial and temporary planning of production. A crop's economic value offers a helpful indication for such planning as it can assist in optimizing earnings and the allocation of water through suitable crop selection [22]. Rice production planning with the aim of minimizing WF means growing rice in provinces where the average water requirement is low [39].

### 3.2.4. Maize

Summer maize is cultivated in all the Iraqi provinces as a feed crop. As shown in Table 1 and Figure 5, the average of the WF per ton was 2238 m$^3$/ton, the green WF average was only 69 m$^3$/ton, the blue WF was the main component with 2169 m$^3$/ton. The green WF of production was only 25,292,837 Mm$^3$/yr (0.27%), the blue was 794,944,139 (99.73%) Mm$^3$/yr and the total WF of production was 820,236,976 Mm$^3$/yr. Babylon, Baghdad, and Wasit have the largest WF. Basra has the largest WF per ton because of the hot climate and low production, Nineveh has the lowest WF (Figures 5 and 6).

### 3.3. Cereals Trade Quantification from 2007 to 2016

The quantities of the four crops Iraq imported from different countries during the 10 years covered by this study from 2007 to 2016 were obtained from FAOSTAT [11] and are shown in Table 2. There was an average of 1,966,256, 6198, 249,212 and 43,826 ton/yr for wheat, barley, rice, and maize, respectively, and the maximum import was for wheat, followed by rice, maize, and barley.

**Table 2.** The amounts of the four crops (ton) imported by Iraq (2007–2016) [11,15].

| Year | Wheat | | Barley | | Rice | | Maize | |
|---|---|---|---|---|---|---|---|---|
| | Production | Import | Production | Import | Production | Import | Production | Import |
| 2007 | 2,202,777 | 2,420,183 | 748,291 | 931 | 392,803 | 92,287 | 384,471 | 9863 |
| 2008 | 1,254,975 | 2,963,320 | 403,999 | 30,211 | 248,157 | 39,026 | 287,955 | 27,253 |
| 2009 | 1,700,390 | 3,050,409 | 501,508 | 23,820 | 173,074 | 30,016 | 238,113 | 10,795 |
| 2010 | 2,748,840 | 1,854,525 | 1,137,169 | 233 | 155,829 | 108,740 | 266,699 | 2840 |
| 2011 | 2,808,900 | 2,888,833 | 820,152 | 2000 | 235,118 | 137,291 | 335,710 | 2068 |
| 2012 | 3,062,311 | 2,425,393 | 831,990 | 0 | 361,339 | 307,012 | 503,389 | 3704 |
| 2013 | 4,178,379 | 2,113,311 | 1,003,198 | 0 | 451,849 | 264,906 | 831,345 | 11,463 |
| 2014 | 5,055,111 | 1,519,185 | 1,277,796 | 4767 | 403,028 | 292,338 | 289,288 | 3367 |
| 2015 | 2,645,061 | 252,272 | 329,713 | 0 | 109,209 | 522,883 | 182,340 | 84,759 |
| 2016 | 3,052,939 | 175,125 | 499,222 | 22 | 181,320 | 697,625 | 259,546 | 282,144 |
| Average | 2,870,968 | 1,966,256 | 755,304 | 6198 | 271,173 | 249,212 | 357,886 | 43,826 |

Water footprint data on crops was essential to quantify the water footprint of trade (WFT) and evaluate its effects on water footprint reducing and land savings, this was calculated by Equations (9)–(11) and showed in Table 3. The annually average quantity of cereals imported by Iraq between 2007 and 2016 was (3,478,306,864), (10,964,262), (821,087,552) and (98,082,588) $m^3$/yr for wheat, barley, rice, and maize respectively. The maximum volume imported was for wheat, followed by rice, maize, and barley, the same order for the 4 crops was to water-saving and land saving (Table 3).

Importing crops could lead to low food self-sufficiency in Iraq, but WFT has the advantages of water and land savings. Iraq's national resource executives and trade policymakers would profit from a stronger knowledge of the connection between global trade and domestic resource conservation, and these findings could provide helpful data for Iraq. Table 3 demonstrates that crop imported water savings in Iraq amounted to 2.6 billion $m^3$/year representing 6% of Iraq's total water withdrawal (42.8 billion $m^3$/yr), and land savings amounted to 1.2 million ha, representing 20% of Iraq's total agricultural land (6 billion ha), indicating that Iraq's crop trade has a great water and land resource advantage (Table 3).

The water footprint of trade is considered a significant problem in importing nations in terms of water and food security. For example, by replacing imported food products with national food products, reducing WFT may be related to water consumption. It should be clear that reducing the WF plays a role in achieving sustainable water, land and food safety, although the definition of that is narrow and difficult to apply [22].

**Table 3.** Average of water footprint of trade (WFT), water-saving, land saving, and the ratio (%) of saved water and land to internal water resources and agricultural land area in Iraq resulting from the four crop amounts imported by Iraq (2007–2016).

| | WFT ($m^3$/yr) | % | Water Saving ($m^3$/yr) | % | Land Saving (ha/yr) | % |
|---|---|---|---|---|---|---|
| Wheat | 3,478,306,864 | 79% | 1,659,520,064 | 62% | 1,156,514 | 93% |
| Barley | 10,964,262 | 0% | 3,879,948 | 0% | 4636 | 1% |
| Rice | 821,087,552 | 19% | 918,097,008 | 34% | 63,600 | 5% |
| Maize | 98,082,588 | 2% | 95,058,594 | 4% | 14,789 | 1% |
| Total | 4,408,441,266 | | 2,676,555,614 | | 1,239,539 | |

Iraq is rapidly facing water scarcity or water stress conditions. Since the 1980s, the water flow of the Euphrates and Tigris rivers has dropped by 30% and is projected to decline further by up to 50% before 2030 [34]. The total water supply is expected to decrease by up to 60% between 2015 and 2025 [3,40]. Iraq's food basket, located in the country's central south, has lost around 50% of its production capacity over the past two decades due to salinization [41].

It is clear that there is water scarcity in Iraq, water scarcity refers to the disparity in water availability over a specific period of time and for a particular region, it is the blue WF ratio to the availability of blue water, changes during the year and throughout the year [18].

The data from the Gravity Recovery and Climate Experiment (GRACE) satellite system can provide estimates of water equivalent variations and assess the regional water availability changes caused by water movement and water depletion across the area studied [42].

Bozorg-Haddad et al. [43] studied the water scarcity in the Middle East region (including Iraq) by Gravity Recovery and Climate Experiment remote-sensing technique, their analysis indicates descending trend of water storage in the region. The two water scarcity indices, water equivalent anomaly (WEA) and total water storage (TWS) depletion calculated and revealed that the region has very negative water scarcity indices. Lebanon has scored the worst annual average WEA ($-14.26$), Syria ($-10.25$), Iraq ($-10.17$), and Iran ($-9.76$). Such results show the need for international cooperation to control the global water resources available and reverse the depletion of natural water sources and this is consistent with the results and recommendations of this study.

Beside the climate change there are internal reasons for Iraq's water challenges, the most important of which can be mentioned [7]:

- Lack of strategic planning in the long term.
- There is no regional cooperation.
- Lack of a development plan for human resources.
- High losses of water through the network of distribution.
- Water scarcity and increasing demand for water.
- The need to build new infrastructure on a large scale with high investment needs.
- Rapid increase in population.
- Lack of agreements with neighboring countries on water distribution.
- Shift in climate, rainfall patterns and rising temperatures in Iraq.

## 4. Conclusions and Recommendations

Generally, there is relatively reduced rain-fed land production and reduced green water chances compared to blue water in Iraq.

The idea is that taking the opportunity to improve the use of green WF (by adjusting crop sowing dates) or to boost the productivity of green WF (by boosting yields in rain-fed land, for example by introducing additional irrigation) can be of great importance to farmers who already have insufficient (blue) water resources [22,44].

Increasing rain-fed land production would minimize the need for a higher rate of irrigated land production in water-scarce areas such as Iraq, resulting in decreased demands for blue water, which have been shown to be big components of the complete water footprint of cereal crops production.

The results indicate that the approach of the government to tackle scarce agricultural water resources should be reconsidered by concentrating exclusively on enhancing irrigation effectiveness. While surface water irrigation systems are expensive and difficult for most farmers to maintain and operate, improving the productivity of water (particularly for green water) in cereal production can be accomplished more readily with improved farm management methods such as altering sowing dates or using new cultivars with a more adjusted, increasing season for moist spells in each region. Any other alternatives leading to enhanced yield and greater productivity of green water are similarly crucial (such as implementing additional irrigation). The information and estimated values presented

in this research should assist policymakers in making more informed choices about allocating restricted agricultural water resources.

Importing cereals could be an effective water portfolio that dominates water management in water-scarce nations like Iraq and can provide fresh perspectives for understanding and addressing water pressure and scarcity in water-deficit regions like Iraq. The quantity of cereal crops imported is considered to be the most important factor in determining water and food security and results from water and land savings in Iraq that may show the importance of international trade. In short, policymakers can profit from taking into account both the quantitative effects of importing cereal crops and WF's structural change, such as fragile growth (or decrease) in Iraq. Cereal crops trade intensity and connectivity analyzed in this research can be a significant element for incorporating food and water policy in Iraq, and this research could provide policymakers with significant data for assessing future resource management scenarios for sustainability in Iraq.

**Author Contributions:** S.H.E. and S.A.A. collected the data, analysis, validation investigation, data curation, and writing—original draft preparation. N.A.-A. did the visualization, project administration, and writing the final draft. All authors have read and agreed to the published version of the manuscript.

**Funding:** This research received no external funding.

**Conflicts of Interest:** The authors declare no conflict of interest.

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
