# Peer review of "Assessment of Main Cereal Crop Trade Impacts on Water and Land Security in Iraq"

_agronomy, doi:10.3390/agronomy10010098_

Round 1

Reviewer 1 Report

I am happy that the specified changes were done and the paper now should be accepted for publication

Author Response

Thank you very much.

Best regards.

Nadhir Al-Ansari

Reviewer 2 Report

Authors provided an interesting manuscript focusing on water footprint and strategies for water saving at national scale. However, Authors stress out the concept of virtual water while their manuscript analyses the water footprint and the water saving by trade. For this reason, authors need to address better their goal and scopes and reshape the discussion part. Please, accept the following suggestions in order to rebuild your manuscript in a better form for publication.

Authors need to consider using the Journal’s English revision. Change the title since virtual water is not addressed on your manuscript unless you consider water footprint and water saving by trade to be like “virtual water”. Syntax need to be revised overall the manuscript Line 80-82: The citation of virtual concept mentions only partially the first concept, while “virtual water” is the water embedded and consumed for a specific crop production traded (imported and exported) by and from a country. You address only the water footprint of import. Line 83-84: no sense sentence. Please rephrase it. The following sentences concerning virtual water need to be revised. A better definition of virtual water needs to be addressed. Line 125: is there any percentage of water shortage in Iraq? Line 160-161: what’s the model by Hoekstra and Hung 2002? Cropwat? By the way, this reference is missing in the list. Authors might consider merging chapter 2.1, 2.2, 2.3 and 2.4 since this division make readers confused and data collection is not well illustrated. Equation 2: what does “lgp” stand for? Write the acronym. Equation 6 and 7: how did you calculated Reff? Chapter 2.6 and 2.7 can be merged in a single “water saving” methodology chapter. Here, using the virtual water concept is misleading; authors can refer to the water footprint reduction while importing food by trade. Chapter 3.2: check the unit of measure (Mm3), you mentioned that at country level only 42.8 km3 are withdrawn. Line 266-268: unit of measures are missing. Authors mixed surface, yield crop or crop production and amount of imports. Line 345-360: This part is not well positioned in the manuscript, it cannot be in the “maize chapter” since authors changed topic in those paragraphs. Chapter 3.3: the “virtual water” is usually calculated as exported minus imported embedded water in food production, whereas import virtual water is made by the virtual water from the exporting countries. In this study, authors involve only national water footprint of the four main crops. Mainly, the study lack of a statistic analysis, as for example a sensitivity analysis would be a right evaluation to understand what variable mostly affects such amount of WF both for green and blue water. Why some areas of Iraq spend more water than others? What does affect the higher green value in a certain location rather than another in Iraq? Conclusions need to be reconsidered and rephrased according to the changes overall the manuscript.

Author Response

Dear Reviewer

Thank you very much for your comments which improved our paper.

Best regards.

Nadhir Al-Ansari

This is our response to the reviewer's comments

1. The study title changed from (Virtual Water of Main Cereals Crops in Iraq) to (Assessment of Main Cereals Crops Trade Impacts on Water and Land Security in Iraq).

2. Some phrases and acronyms in the abstract and all the study have been modified to match the new title.

3.  Line 51: The (CSO) is defined and (2019) is deleted.

4. Line 70: The (fourth) changed to (the third most crop).

5. Line 80-82: All the paragraph 80-88 is rephrased.

6. Line 91: (WF) and (VW) are deleted.

7. Line 100 and line 101: the word (need) is added after (water-intensive).

8. Line 104: (VW) changed to (WF).

9. Line 125: The percentage of water shortage in Iraq was mentioned clearly in the 47-50 lines.

10. Line 130: The (VW trade) changed to (cereals trade to Iraq).

11. The chapters 2.1, 2.2, 2.3 and 2.4 are merged, now within the title: 2.1 Data.

12. Line 141: The (4.28 Km3) changed to (42.8 billion m3/yr).

13. Line 160-161: Corrected, Hoekstra and Hung method in the [36] reference and the CROPWAT reference is [38] in the list.

14. Equation 2: This phrase was added: (Where lgp is the duration of the period of growth).

15. Equation 6 and 7: The answer for the question (How did you calculated Reff?) is: (The rain data in CLIMWAT was used to calculate the effective rainfall (Eff. rain) using the United States Department of Agriculture (USDA) S.C. method [Allen, 1998].

16. Chapter 2.6 and 2.7 merged under the title:

        2.3 Water and land saving methodology

17. Line 266-268: corrected (The phrase was removed).

18. Line 340-360: The paragraph is now corrected and separated under the title:

         3.3 The importance of studying the agricultural water footprint

19. Line 364: (2019) was deleted.

20. Chapter 3.3 (now, Chapter 3.4) and the title corrected to:

     (3.4 Cereals trade quantification to Iraq from 2007 to 2016).

     -Because Iraq is a food importer and exports little at times.

     -Because of the difference in climate between southern and northern Iraq, in the north the rains are more, the temperature is lower and the land is mountainous and cannot be irrigated from rivers, so dependence on green water will be. In contrast, in the hot south and less rain, the consumption of blue water is more.

21. The discussion part reshaped as possible.

22. The English language is corrected.

23. Line 390: The conclusion is corrected and paraphrased now and the title becomes:

    (4. Conclusions and recommendations), some phrases and acronyms have been modified.

Round 2

Reviewer 2 Report

After major changes, authors need to address better their goal and scopes and reshape the discussion part, which is now in a single “results and discussion” section. Please, accept the following suggestions in order to rebuild your manuscript in a better form for publication.

GENERAL COMMENTS

The English syntax needs to be revised. Authors considered only WF assessment in their analysis, while a “blue water scarcity” analysis can be useful for readers to understand the pressure on water resource of agriculture in Iraq (42.8 km3 in relation to the blue water from crop production) and for authors supporting their thesis of using food trade for save water (change of blue water scarce according to different food trading or different areas of Iraq). You can refer to the “Water footprint manual” by Hoekstra et al., 2011. A discussion section would better address your final remarks and general considerations of your main outcomes, besides merging your results in a single long and not well structured “results and discussion” section. I recommend to split “results and discussion” in 2 different sections.

SPECIFIC COMMENTS

Line 135: study area in brackets is not needed. Authors might rename chapter 2.1 as “Study area and data collection”. Chapter 2.3: authors changed title, while the concept of virtual water trade is still misleading, because they only accepted part of the concept and calculated importing virtual water simply multiplying the national crop product by the internal water consumption. Authors are still analyzing WF saving! I recommend using the term of WF due to its broader scope. Line 156: the method Hoekstra and Hung used for plant evapotranspiration is the same as in Cropwat, which is addressed in Book 56 of FAO. Hoekstra and Hung only related the CWR with yield or surface. Please readdress the citation in a correct form. Figure 5 and figure 6 would be placed in a better position, maybe after chapter 3.2? Or you can consider placing every graph for single crop. Chapter 3.3: the title does not well address the concept of this paragraph, which is general and not supported with any result. Please, add more outcomes or consider replacing this part in a discussion chapter.

Author Response

Dear Reviewer

I would like to thank you very much for your valuable comments and efforts that improved our paper. Please note that we did what is required. I hope that the paper is satisfactory now.

Thank you again.

Best regards.

Nadhir Al-Ansari

Response to the reviewer's comments

1. The English syntax had been revised. 

2. The discussion part is reshaped and new paragraphs added to enrich it.

3. The “blue water scarcity” was referred to and discussed regarding Iraq in the last part of the discussion part.

4. The terms (water footprint of trade) or (cereal trade) were used in place of the (virtual water) or (virtual water trade) in the appropriate places of the study.

5. New paragraphs were added to the discussion of the study.

6. Line 145: The words in brackets is deleted. The chapter 2.1 is renamed as “Study area and data collection”.

7. Chapter 2.3: The term (WF) is used instead of (WF) where required.

8. Line 156: The citation is corrected.

9.Figure 5 and figure 6 are placed after chapter 3.2.

10. Chapter 3.3: the title is deleted and all this paragraph was moved to a more suitable place before the paragraph (3.2.1 Wheat).  

This manuscript is a resubmission of an earlier submission. The following is a list of the peer review reports and author responses from that submission.

Round 1

Reviewer 1 Report

Title                 Virtual Water of Main Cereals Crops in Iraq

Authors:         Salam Hussein Ewaid , Salwan Ali Abed , Nadhir Al-Ansari

1. The introduction would need some further analysis regarding virtual water, and water sustainability in agricultural productions.

2. The paper relates to the years 2007 to 2016: would it be possible to add values also from 2017 (and/or 2018 or 2019) to increase the robustness of the approach?

3. Table 1 and 2: please indicate homogeneous number of decimals between values in the same column Table 2 values are not reasonable: the number of significant digits (in particular in the Total Production) is not reasonable with the uncertainty of the values themselves: please use scientific notation (e.g. ×106)

4. It would be important to add estimation of grey water footprint in the analysis.

5. A revision of English language and style is needed.

6. Section 2.4 Agricultural and trade data- please address the following:

        a. More detail is needed here - what data is available? What other sources could be used? How long has the data been collected? Is the data publicly available?

        b. I am not an expert on CROPWAT so more information and details on what USDA conversion means and why it is important

7. In the Conclusions, could the recommendations and applications be developed a bit further? What applications exactly could the virtual water have?

Author Response

Respond to the comments of the first reviewer, requesting acceptance

Dear Reviewer

We would like to thank you very much for your comments which improved our paper.

Below is the response to your comments.

Thank you again.

Best regards.

The introduction is further analysis now, required was done Data for 2017, 2018 and 2019 cannot be added because statistics are not available from their sources. In Table 1 and 2: the decimals between values are added. Grey water footprint cannot be studied because of the lack of data. English has been improved as much as possible.

Section 2.4 Agricultural and trade data:

        The data used in this study included:

The website of the Iraqi Ministry of Planning provides statistical data publicly on the production quantities of crops and cultivated areas for previous years (in Arabic). http://www.cosit.gov.iq/en/ The Iraqi Meteorological Organization and Seismology (IMOS) provide weather data on the Iraqi metrological station's for previous years, (in Arabic), unpublished data. http://www.meteoseism.gov.iq/ The Iraqi Ministry of Agriculture, Department of Extension and Agricultural Training, Baghdad, Iraq. Provides data about crops (cultivation and harvesting times) within the Guide to Agricultural Operations in Iraq (in Arabic), unpublished data. The FAO website provides data on crop production and climate in the world obtained by the CROPWAT and CLIMWAT software. Conclusions

Calculating the water footprint help to know the source, time and quantity of water needed to grow a crop or produce a commodity and have many applications and benefits, including combating drought and water shortages and replacing commodities and crops that require much water. It is an important tool within the proper integrated management of water resources, and that was mentioned in this study.

Reviewer 2 Report

The mauscript "Virtual water of main cereal crops in Iraq" tries to perform a  water footprint assessment of crop production in Iraq using the software CROPWAT.

In my view, this manuscript does not add any results to the current literature in the filed of water footprint and virtual water trade. Importantly, the text is not written in a proper English: many sentences come without a verb. The introduction is just a list of sentences without any connections and the main research question/goal of the study is stated in 2 lines without any links to current gaps in the literature. The overall Introduction needs improvement to better clarify current literature, what is missing, and whar are the main goals of your paper. Results are order of magnitude larger than typical WF results. For example, the authors found a WF of 2*10^15 m3/year for wheat production in Iraq. But we know from Mekonnen and Hoekstra (HESS, 2011) that the global WF of all agriculture is around 7*10^12 m3/yr. I would suggest to compare your results with previous studies. Moreover, I would suggest to improve figures. Results cannot be just a collection of histograms. What about providing maps to show the sub-national variability of crop WF or the virtual water fluxes?

I have attached some comments/suggestions I have put along the text. Please follow them if you think they can help you to improve the manuscript.

In my opinion, this study is not suitable for publication on Agronomy.

Author Response

Respond to the comments of the second reviewer, requesting acceptance

Dear Reviewer

We would like to thank you very much for your comments which improved our paper.

Below is the response to your comments.

Thank you again.

Best regards.

All the notes in the PDF file are responded to in detail as in the manuscript. English has been improved as much as possible. The introduction has been completely reformulated. The numbers seem large because they belong to the production of the four important crops in all of Iraq and what consumed of rain water and rivers water for a year. The other references are even if I talked about Iraq is theoretical approximations and other periods have been confirmed by contacting its authors.

Reviewer 3 Report

This paper is a potentially very interesting case study of an arid country (Iraq) that has been utilizing virtual water imports as an essential water management strategy.  As population grows, this will be ever more essential. In its current form, however, the paper needs substantial improvement, bith substantively and in presentation. Below are specific points:

1. The use of English is only fair and needs substantial improvement.

2. The use of cartography would greatly improve the presentation. Replace Figure 1 with a map of specific crop-growing areas in Iraq, perhaps with rainfall totals in the background. Fig 6 graphs would be also better presented as maps, perhaps with blue and green bars emerging from Iraq's provinces.

3. FAO has developed AQUACROP; does this make CROPWAT obsolete? 

4.Fig 4: Y-axis is unclear.

5. Can you compare the water footprint of imported crops with their water footprint in Iraq?  This would tell us whether virtual water saves actual green and especially blue water system-wide.

6. Broadly, the argument that virtual water imports is an essential water management strategy for Iraq is very important. As written, however, the paper only glances at this argument. What are the limitations on domestic water resources, especially the Tigris and Euphrates Rivers? Are these now being approached or exceeded? Is this why Iraq has been rapidly increasing imports of wheat, maize and rice? Will ever-more imports be required as the population increases? How much will the population increase? Is it assumed that all food imports replace irrigated rather than rainfed agriculture?

In other words, you need to paint a picture of how much worse the domestic water situation would be if Iraq did not import virtual water through food and how virtual water abates that crisis. 

Author Response

Respond to the comments of the third reviewer, requesting acceptance

Dear Reviewer

We would like to thank you very much for your comments which improved our paper.

Below is the response to your comments.

Thank you again.

Best regards.

English has been improved as much as possible, especially in the introduction. Maps cannot be made due to lack of experience. Each program has different uses,

AquaCrop is particularly suited to address conditions where water is a key limiting factor in crop production.

CropWat is a decision support tool for the calculation of crop water requirements and the development of irrigation schedules for different management conditions.

Fig 4 replaced with clear one. Unfortunately, there are no other Iraqi studies on the water footprint. Integrated water management can provide the solutions for reducing WF like shifting from producing and consuming higher water using products to less water using products. The integrated water management, understanding the product's WF helps to promote improvements in consumption patterns and implement water-saving behavior. The water footprint of an area can be reduced by several ways, the first way is approving production that needs less water per unit and gives more harvests per unit of water used. For example, applying up-to-date technology in irrigation and improving the rain water use in additional irrigation. The second way is to switch from consuming extremely water-intensive goods to products that are less water-intensive, such as reducing meat consumption. A third way to reduce WF is to shift the production from low water-productivity countries to high water-productivity countries. Jordan is an example where it is externalizing its WF by depending on imports of wheat and rice from the higher water productivity countries like USA.
